# Decisions and choices about fertility and family planning: Perspectives from husbands and wives in Sudan

Dina Badri[1¤]*, Anja Krumeich[2], Bart Van de Borne[3†]

1 Department of Organisational Management, Ahfad University for Women, Omdurman, Sudan,
2 Department of Health, Ethics and Society, Maastricht University, CAPHRI, Maastricht, The Netherlands,
3 Department of Health Promotion, Maastricht University, CAPHRI, Maastricht, The Netherlands

¤ Current address: 10441 Spring Green Blvd, Katy, Texas, 77494.
† Deceased

* badrid21@gmail.com

## Abstract

### Background

The use of family planning contraceptives is an essential feature of reproductive health and rights, and safeguards women's health. Sudan is among the countries with the lowest prevalence of family planning use in Africa. The social construction of gender norms related to preference for increased fertility and family planning decisions resides in the assumption that husbands play a limiting role in their wives' access to family planning use. This study aims to examine to what extent husbands' preference for increased fertility is reflected in their wives' contraceptive use and whether and how their wives adopt their wishes in their actual family planning use.

### Methods

This study used a qualitative approach. Khartoum and Omdurman urban cities were included in the study because they demonstrate high fertility, low contraceptive use among women, and increased unmet need. Individual interviews were conducted with forty-six participants (husbands and wives) in Khartoum and Omdurman cities. The Fertility and Reproductive Health Services Centre (FRHC) in Khartoum and Ahfad Family Health Centre (AFHC) in Omdurman were suitable for this study because of their many years of providing reproductive health services. The medical directors and key health providers assisted in identifying potential participants in the health center's catchment areas. The participants, the husbands and their wives, were interviewed in separate spaces at FRHC, AFHC, in their homes, or at their workplace. Responses from participants were audio-recorded, transcribed, translated from Arabic to English, and thematically analyzed.

**Data availability statement:** All relevant data are within the manuscript and its Supporting Information files.

**Funding:** The author(s) received no specific funding for this work.

**Competing interests:** The authors have declared that no competing interests exist.

## Results

The findings reveal that husbands' preference for increased fertility does influence their wives' contraceptive use. Wives who had different perceptions and were concerned about their reproductive health decided not to conform to spousal influence, seeking means to address their contraceptive needs. This is summarised as follows: *husband and wife agree about fertility and the use of family planning; *husband's preference for increased fertility overrules wife's considerations of contraceptive use; *some space for women to decide and use family planning methods; and *wife's decisions to use family planning methods.

## Conclusion

This study's findings posit the call for partnerships forged between national health officials at the policy level, public health researchers, community workers, and community leaders to acknowledge the importance of men's involvement in family planning. Conducting further studies and initiating awareness-raising programs focusing on men's contraceptive literacy and, hence, attitudes towards family planning, can reduce their influence in reproductive health decisions and promote their role as supporters for women's contraceptive use. Furthermore, national reproductive health policies must recognize the critical role health providers can play in engaging men. Implementing training initiatives for health providers to improve their roles in counseling men about the health benefits of family planning use can lead to increasing couples' discussions about fertility-related behaviours to better safeguard their reproductive health status.

## Introduction

The use of family planning methods is a crucial aspect of safeguarding reproductive health, a key strategy in improving maternal health, averting and reducing unintended pregnancies, and contributing to lowering fertility rates. As per the International Conference on Population and Development (ICPD) and the United Nations Population Fund, family planning is a right that ensures unrestricted access to contraceptive services and information, and the freedom to decide the number and spacing of children [1–3], thereby regulating fertility. Despite this, women's utilization of family planning remains low in many parts of sub-Saharan Africa, a region known for its high fertility rates and considerable contraceptive unmet need [4–6].

Sudan is a high-fertility country with a rate of 5 births per woman [7,8], which has rapidly increased its population. According to reports, the country's population growth will double to 80 million from its current population of approximately 50 million [9]. It has been characterized as one of the countries with the highest rates of unmet need for family planning within the Middle East and North African region, increasing from 18 percent in 2022 [10,11] to only 29 percent in 2023.

Sudan remains to have one of the lowest contraceptive prevalence rates in the sub-Saharan African region at 9% [9,10]. Studies from Sudan report that about one in

every ten married women relies on pills. Less than 1 percent use natural methods [10], leading to a high estimated number of 390.000 unintended pregnancies [10]. Sudan has increasing rates of maternal mortality, estimated at 295 deaths per 100,000 live births [8,12], and a high rate of infant mortality at 38 per 1000 live births [9,12]. Women's family planning use and reproductive health status are further exacerbated by a prolonged history of wars, destroying the country's health infrastructure [13–15].

Low levels of contraceptive use are also associated with women's low socioeconomic status [16] and myths about long-term contraceptive use, which are believed to reduce women's reproductive roles [17]. Studies from Sudan confirm that normative beliefs and cultural traditions favouring increased childbearing among women have a profound impact on women's low use of family planning methods [16–19]. Traditions that encourage early marriages and the expectation to conceive soon after being wed [20] glorify a woman's social status. Therefore, deferring contraceptive use implies a lack of self-agency in contraceptive decision-making. Similar findings have been reported in Tanzania and Somalia, showing that women's contraceptive agency is minimised by the value and social status of womanhood [21,22].

In cultures that highly value increased childbearing, many parts of sub-Saharan Africa have shown that men's desire for large families is more commonly reflected, contributing to rising fertility rates [23–26]. Studies from Ghana [27], Uganda [28], and Senegal [29] report that a man's status and honour are measured by having a large family, and wives are married to become child bearers [28]. In Sudan, men's increased fertility-related behaviours have been reported [30]. The belief that children are future providers for the family, the importance of having a son(s), the desire to continue the father's lineage, and the growth of the family clan are primarily guided by cultural traditions and socio-religious norms [20,30–32].

In this regard, multiple studies in sub-Saharan Africa have realised that the interrelation between men's demand for increased childbearing and their traditional role as head of the household and primary breadwinner increases their position to influence contraceptive decisions, by way of controlling a woman's contraceptive use [5,33–37]. Other studies from Uganda [38,39] and Nigeria [40] report that men's opposition towards a woman's contraceptive use is largely a result of their misconceptions about the use of family planning methods, believing it will adversely affect women's fertility; therefore, strengthening their control over use is seen as justified.

In Sudan, whilst studies have shown factors and behaviours related to women's use and non-use of family planning methods, mainly collecting data from women only [16–19,41], much less has been examined about the interrelation of women's contraceptive use and men's fertility-related preferences.

This study aims to examine the extent to which husbands' desires for increased fertility are reflected in their wives' contraceptive use and whether and how their wives adopt their wishes in their actual family planning use.

Findings from this study can provide rigorous scientific data to inform national policies and future research about the need to engage men in family planning programs, whilst recognizing the centrality of women's agency to make contraceptive decisions for improving their reproductive health status.

The terms 'family planning use' and 'contraceptive use' are used synonymously in this study.

## Methods

### Study setting

This study was conducted in Khartoum, the capital of Sudan, and in Omdurman, west of Khartoum, with a total population estimated at 6.5 million [42], representing people of different Arab and African ethnic backgrounds with a predominantly Muslim population of approximately eighty percent and a Christian population of roughly twenty percent [12]. The cities rely on the agricultural way of living (farming, livestock, and fishing), which employs almost 80% of the Sudanese and makes up a third of the economy. The urban cities of Khartoum and Omdurman were selected because they represent high fertility, low contraceptive use among women, and increased unmet need despite the presence of the Sudan National Family Planning Centre in Khartoum [10,16].

 

## Study design

This study used a qualitative design. The health centres included in the study were the Fertility and Reproductive Health Services Centre (FRHC) in Khartoum and the Ahfad Family Health Centre (AFHC) in Omdurman. They are centrally located in Khartoum and Omdurman and have provided comprehensive reproductive health services to the communities.

Before the data collection phase, the leading researcher approached the medical directors and key health service providers from both health centres to share and discuss the study objectives. The medical directors and key health services providers have extensive experience providing family planning and reproductive health services in their respective communities and were, therefore, suitable to assist in identifying entry points and potential study participants in the health centres' catchment areas.

## Participant recruitment and inclusion criteria

Participants were selected using purposeful sampling [43] from the communities served by the FRHC in Khartoum and the AFHC in Omdurman. Using a snowballing approach, the medical directors and key health services providers asked potential study participants to refer further participants. This approach was relevant for this study because it allowed for developing a sense of trust and comfort among the husbands to come forward and voluntarily participate in the interviews [44]. It is noteworthy to mention that in this study context, a gender data gap is evident. Men have hardly been part of previous data collection regarding fertility and family planning in the scientific literature. Furthermore, to counter any possible bias during the sampling process [44] and ensure methodological rigour, the study includes participants representing diverse socio-economic and religious backgrounds, different age ranges, varying parity, and a fairly balanced sample representing Omdurman and Khartoum cities (Table 1).

The inclusion criteria for the participants were married couples with children, residents of Khartoum or Omdurman, use of family planning methods (modern/natural) at the time of the study, and willingness of both the husband and his wife to participate. The exclusion criteria were participants' unwillingness to partake in the study.

## Data collection and study procedure

The data was collected from December 2017 to January 2019 using semi-structured interview questions by the leading researcher from Ahfad University for Women and the data collectors' team (five males and four females) from the Community Animators Friendly Association (CAFA), a community-based organisation in Omdurman, and the Sudan Family Planning Association (SFPA) in Khartoum. Collaborating with data collectors from CAFA and SFPA organisations was beneficial due to their experiences in research and data collection within the community and extensive knowledge about reproductive health and family planning. The leading researcher convened several meetings with data collectors before the commencement of field work to explain and provide specific information about the study's objectives pertaining to concepts of fertility behaviours and contraceptive use. The study guide was reviewed by the data collectors' team to ensure clarity and comprehension (S1 Appendix). Further meetings continued during fieldwork to discuss any emerging themes or inconsistencies to facilitate data collection.

For this study, there was no prior limit to the number of interviews. Interviews continued until saturation was observed, at which point no new or important insights were identified for the explored concepts [45]. Accordingly, the study included interviews with 46 participants (husbands and their wives). Obtaining data from the men and their wives provided a rich source of information and more nuanced explanations regarding the power dynamics within contractive decision-making.

Verbal consent was obtained from all the study participants. Permission was also obtained to audio record interviews using a digital recorder, since hand recording may be problematic and may not catch each word during interviews. The interviews were then typewritten after each interview.

**Table 1. Socio-demographic characteristics of the couples (husbands and their wives).**

| Variables | Parameters | Frequency (n) | % |
|---|---|---|---|
| Age range | 16-20 | 4 | 8.6% |
| | 21-30 | 7 | 15.2% |
| | 31-40 | 13 | 28.2% |
| | 41-50 | 17 | 36.9% |
| | 51-57 | 5 | 10.8% |
| Type of marriage | Monogamous | 22 | 47.8% |
| | Polygamous | 1 | 2.1% |
| Parity of the couple's | One child | 2 | 4.3% |
| | Two children | 4 | 8.7% |
| | Three children | 6 | 13% |
| | Four children or more | 11 | 23.9% |
| Level of husband's education | No education | 1 | 2.1% |
| | Primary | 6 | 13% |
| | Secondary | 4 | 8.6% |
| | Tertiary | 12 | 26% |
| Level of wife's education | No education | 1 | 2.1% |
| | Primary | 4 | 8.69% |
| | Secondary | 5 | 10.86% |
| | Tertiary | 13 | 28.2% |
| Level of the couple's socio-economic status | Very low (IDP) | 5 | 10.86% |
| | Low | 6 | 13% |
| | Middle | 7 | 15.2% |
| | High | 5 | 10.86% |
| Area of the couple's residence | Khartoum | 11 | 23.9% |
| | Omdurman | 12 | 26% |
| The couple's religion | Muslim | 20 | 43.4% |
| | Christian | 3 | 6.5% |

IDP- Internally Displaced Person (people mainly from Darfur, the western region of Sudan, who have been displaced due to internal conflict, drought, or other poor living conditions).

Semi-structured interviews were conducted in Arabic (the primary language in Sudan) and lasted approximately 60–90 minutes. Before each interview, data collectors explained the study's nature and purpose to the participants. All eligible participants were assured of anonymity.

The study paid close attention to several factors to facilitate data collection and comfort in discussions during interviews. For many Sudanese men, discussing marital matters and family planning is sensitive and usually considered private, and male participants are more comfortable with same-gender data collectors [46]. Acknowledging this, five male data collectors interviewed the husbands and four female data collectors interviewed their wives. To ensure confidentiality and avoid factors that may shape or influence spousal responses [47, 48], interviews with husbands and their wives were always conducted in separate spaces so as not to hear the other's response or observe the ways they responded. As a result, each spouse was unaware of the responses given by the other. The date and place of the interviews were left at the discretion of the participants [49]. Accordingly, fifteen couples were interviewed at their homes, six at their workplaces, and two couples at AFHC and FRHC, respectively.

## Interview guide

Drawing on literature about the roles of men and women in contraceptive decisions in Sub-Saharan Africa and other similar contexts, interview questions were conceptualized and designed to specifically address the Sudanese context [16,18,28,29,35].

This study's interview guide was divided into two sections. The first section elicited the socio-demographic background about the couple from the husband only, since he is the head of the household (Table 1). The second section examined husband-and-wife behaviours of increased fertility, gender power dynamics influencing husband-and-wife family planning use, conditions when husbands accept their wives' use, and the wives' decisions and actual contraceptive use (Table 2). Interviews also obtained information about the couple's use of family planning methods (Table 3).

**Table 2. Summary of pathways generated from the interviews.**

| Themes | Sub-themes summary |
|---|---|
| Pathway 1: Couples responses indicating agreement about increased fertility and reduced family planning use. | * Couples agree to the socio-cultural tradition of increased fertility behaviours. |
| | * Couples agree to reduce or eliminate the use of contraceptives. |
| | * Couples agree and enact the perception that husbands are the primary decision makers in a wife's contraceptive use. |
| Pathway 2: Couples responses indicating husband's preference for increased fertility overrules a wife's considerations of contraceptive use. | * Couples disagree about family size and contraceptive use. |
| | * Husbands' contraceptive decisions lead to wives' unmet need & wives having to comply with their decisions. |
| | * Gender inequity in contraceptive decisions. |
| | * Husbands' misconceptions about the use of family planning will affect the wife's fertility & and the wife's understanding about the use of family planning will lead to improving her reproductive health condition. |
| Pathway 3: Couples responses indicating husbands allowing some space for wives to decide about contraceptive use. | * Husbands' misconceptions that modern methods limit fertility, therefore allowing the wives to use only natural contraceptives. |
| | * Wives are restricted to using natural methods, but prefer modern methods because they are more reliable. |
| Pathway 4: Wife's decisions concerning her contraceptive use. | * Wife's vast knowledge of modern and traditional contraceptive methods allows them to make their own choices. |
| | * Wives' access and presence at family planning centres alone create more freedom to choose their desired methods. |
| | * Wife's report that family planning services are their own space. |
| | * Wives report satisfaction with family planning providers. |
| | * Wives mention that protecting their reproductive health and well-being is their responsibility. |
| | * Wives make decisions to use LAC methods covertly. |
| | * Wives in advanced educational and employment status perceive that only women should make decisions about contraceptive use and family size. |
| | * Wives pursuing careers increase their contraceptive use. |

**Table 3. Married couples' ever use of family planning methods (46 participants).**

| Variable | Type of method | Frequency | % |
|---|---|---|---|
| Wife's contraceptive use | Modern method (pills & LAC) | 10 | 21.7% |
| | Natural method (breastfeeding only) | 4 | 8.69% |
| | No method | 11 | 23.9% |
| Husband's contraceptive use | Male method | 0 | 0% |

## Data analysis

All interview transcripts were carefully read and translated into English. The transcription of all data was systematically analysed through identifying keywords, selecting codes to interpret and develop themes, categorizing and refining emerging themes into sub-themes in line with the study aim. This systematic approach of analysis helped to maintain a clear interpretation of the intrinsic meanings of keywords found in the data [50]. For further investigation and research, NVivo 11, a qualitative software program, was used [51]. The inclusion of study participants' quotes in the study further illustrated study themes and communicated results.

## Ethical considerations

Ethical approval was obtained from the ethical review and research board of Ahfad University for Women (AUW), the principal researcher's university in Omdurman. Local permission for the fieldwork was obtained from the two participating health centres: AFHC in Omdurman and FRHC in Khartoum.

## Trustworthiness of the study

The theoretical underpinnings of this study were guided by the use of inductive theoretical concepts based on the existing study data, which described relationships of the social phenomenon being explored and developed pathways of inquiry [52,53]. The triangulation of research data between the leading researcher and co-authors guided the exploration of the study structure, enhancing the conceptualisation of study phenomena and ensuring the trustworthiness of findings [54]. The leading researcher convened meetings with key health service providers and data collectors during fieldwork. Frequent meetings with data collectors were held to review work progress, discuss preliminary findings, and address any emerging themes or inconsistencies that could improve subsequent data collection efforts. This was important to validate the data generated by research questions [55]. Continuous recruitment of participants was conducted until data saturation was observed and no new data emerged in response to the main study aim [45]. Study participants represent diversity in their socio-economic, educational, and religious backgrounds to ensure sufficient representation and variation of data.

This study is part of a multistage qualitative research study investigating fertility and family planning among Sudanese husbands in Sudan.

# Results

The socio-demographics show the profiles of 23 couples of husbands and their wives (46 participants). All the couples were in monogamous marriages, except for one, which was a polygamous marriage. All the couples were educated except one. The couples represent variation in socio-economic and living standards, with the largest group from low standards. Many of the wives were housewives and not engaged in paid employment. Most couples were Muslim, the most prevalent religion, and the second most prevalent religion is Christian [9,16]. Many participants were found to be between the ages of 40 and 50, increasing with age, indicating longer marital duration and therefore increased parity. Many had a high parity of four or more children, similar to Sudan's estimated total fertility rate [7, 8], compared to a smaller number of couples with two or fewer children (Table 1).

Four major pathways (themes) emerged from the study data, including: Husband and wife agree about fertility and the use of family planning; husband's preference for increased fertility overrules wife's considerations of contraceptive use; some space for women to decide and use family planning methods; and wife's decisions to use family planning methods. Table 2 summarises these four pathways.

## Pathway 1: Husband and wife agree about fertility and family planning

Husband and wife both adhered to what they saw as childbearing socio-cultural norms and agreed about fertility and the limited use of the wife's family planning contraceptives.

The use of contraception, as expressed by the husbands, are not seen to be patronage to family planning use since it will hinder three main factors, expressing his masculine role through increased virility and fertility, upholding his social religious tradition of fathering large families since children signify social wealth and consolidating his position as head of the household where his decisions are followed and obeyed by the wife, undoubtedly.

The wives of these husbands, however, not only shared their husbands' notions concerning family size but also explained how, in this context, it is not uncommon for them to approve of their husbands' fertility decisions. Adhering to their preference of motherhood and the social moral conduct of wives adopting their husbands' decisions out of respect and recognising their position as primary decision maker suggests, therefore, an important determining factor influencing a wife's contraceptive use as well as an increased family size.

Wife: "*My husband is the head of this household, and wives agree with their husbands' desire to have many children. Also, wanting to have many children is a good blessing.*' Husband: '*I want to have many children. It is important to have many children because it is expected in this culture for men to be able to raise large families. For men, it is seen as social prestige and shows that a man is powerful and expresses his virility*". (Muslim couple, monogamous, three children, Khartoum).

Wife: "*Childbearing for women is important, and I want to have more children, especially since this is the desire of my husband. In our culture, a wife listens and obeys her husband's decisions. Even if I want to use family planning, my husband must agree before attempting to use it*". Husband: "*The traditions and religious values of having large families in this community are the main motivation and reason for men to continue childbearing. Things like family planning are not part of our culture, and that's why I, as the main leader of this household, decide for my wife if she can use contraception or not. A wife's role is to abide by her husband's decisions*". (Muslim couple, monogamous, five children, Omdurman).

### Pathway 2: Husband's preference for increased fertility overrules wife's considerations of contraceptive use

Responses from the couple constitute differing opinions about the wife's family planning use. The husband's perception that marriage is oriented towards increased procreation often precedes their wife's practice of using contraception. Their conviction that contraceptive use must be avoided so as not to hamper their wife's fertility is a clear statement of gender inequity as it relates to contraceptive decisions.

For the wives who were ready to use family planning methods, acknowledging the physical burden of multiple child-bearing, they were compelled to follow their husbands' decisions, deferring their choices of fertility to their husbands' preferences. Wives' discussions with their husbands proved unfruitful, leading to their unmet needs and sustaining a larger-than-desired family size. Neither their contraceptive knowledge nor their readiness to use their desired methods was seen as enough to overcome a husband's opposition. Like in the pathway described in the previous section, the impact of husbands on the couple's use of family planning is considerable.

Wife: "*Having a pregnancy after another is not easy, and I was ready to start using contraceptives, but he did not accept*". Husband: "*Simply…. she needs to avoid taking any methods that will prevent pregnancy*". (Muslim couple, monogamous, six children, Khartoum).

Wife: "*I wanted to use the contraceptive pill after my last child, but my husband refused. He says we only have three children, and he wants many more*". Husband: "*For us men, having large families is a blessing and a duty. We must continue fathering many children because this is our tradition, and men marry to fulfil this duty*". (Muslim couple, monogamous, three children, Omdurman).

## Pathway 3: Some space for women to decide and use family planning methods

Husbands' discussions allowed some wives space to use family planning methods, which may have relieved them from their multiple childbearing roles. A husband's meaningful changes in their perception of family planning methods were typically seen within the prism of using natural methods only. However, it was not a wife's preferred or suitable choice. A husband's misconceptions that the use of family planning will limit childbirth, as well as their social responsibility to prevent their wives' contraceptive behaviour, prompted him to accept and approve of a less effective and unreliable method. This could therefore be a more subtle way of denying her access to her desired contraceptive choice, allowing the possibility of a soon-to-be pregnancy and sustaining the couple's already large family size.

> Wife: "*He agreed I could use only natural contraceptives after my last birth. I used breastfeeding for a short time, and soon I was pregnant again. However, I wanted to use another type, like the pill, because it is effective and safe*". Husband: "*Family planning is a means of limiting childbirth, and in this community, it is not welcome. When my wife wanted to use family planning, I told her she could only try natural methods for some time*". (Muslim couple, monogamous, six children, Omdurman).

## Pathway 4: Wife's decisions to use family planning methods

In contrast with the pathways detailed in previous sections, some wives reported that family planning is a woman's choice and decision. Their self-efficacy of not aligning with their husband's influences and dominant socio-cultural norms was associated with increased impact on their actual contraceptive use. Several factors have confirmed this, such as the wives' presence and access to family planning centres without their husbands, the nature of the family planning structures responding to women's needs, advanced socio-economic statuses of wives, and the understanding that a women's body and her health is her prime responsibility were reported as critical to transform their realities and make informed contraceptive decisions as a means to control their family size and reduce their contraceptive unmet need.

Table 3 shows that wives who used any methods preferred the use of pills and long-acting methods more than relying on natural methods. In contrast, none of their husbands reported using any family planning methods.

The wives' knowledge and use of modern or natural family planning methods were driven by the importance of birth spacing, which was seen as equally important to guard against the impact of multiple pregnancies on their well-being and allow them sufficient time to recuperate.

> "*I know about the different methods. Generally, married women are well-informed about their contraceptive methods and can choose any method they desire*". (Muslim wife, monogamous, one child, Omdurman).

> "*Use of family planning is significant for a mother's health. Contraceptive use gives us time to relax between pregnancies. I know about different modern contraceptives, but I generally rely on breastfeeding for spacing*". (Muslim wife, monogamous, seven children, Omdurman).

As some of the wives indicated, they were very upfront about their opinions that family planning programs concern only women and, therefore, their independent space. Their husbands' lack of family planning knowledge or involvement was nevertheless seen as a factor that discredits and disempowers a husband's role in intervening with their access to family planning services freely.

> "*He doesn't know about family planning. Family planning is a woman's affair and does not concern them*". (Muslim wife, monogamous, four children, Khartoum).

Wives showed no interest or readiness to invest in engaging their husbands during the consults with the family planning providers.

One wife said:

*"Usually, only wives go to the family planning centre for their needs; generally, men don't go with their wives, and I never thought of taking him"*. (Muslim wife, monogamous, two children, Omdurman).

For the wives, the importance of taking care of their well-being and finding means to support their health, as well as their perception that family planning services are accessible to obtain their contraceptive methods, were enabling factors in their endeavour to make their own decisions about which type to use and where to receive them.

*"It is a woman's responsibility to care for her health. I am the one who endures all the body changes of pregnancy and even sometimes illnesses from pregnancy by myself. So yes, women need to be in charge of their health. I went to the family planning centre not far from here, and I chose to use the pill"*. (Christian wife, monogamous, one child, Khartoum).

Other wives who decided to use contraceptives reported satisfaction regarding social interactions at family planning centres. Their presence at family planning centres alone was also perceived as an opportunity to discuss freely with health providers, advising them about their preferred method of choice.

*"I needed to use contraceptives, so I went to the family planning centre alone, and the doctor gave me the pill, which I used for some time. I was comfortable using it because it is important for me"*. (Christian wife, monogamous, five children, Khartoum).

Once again, the husband's lack of family planning knowledge and the implications of their husband's opposition to their contraceptive use, in some cases, led the wife to make decisions to use concealed long-acting methods (LAC), in this case, the injections.

*"I know my husband does not know or agree to family planning. I did not tell him when I used the contraceptive injection"*. (Muslim wife, monogamous, seven children, Omdurman).

Some wives believed their need for contraceptives was much greater than a husband's threat of a possible marital dispute. Spousal discussions over the wife's contraceptive use indicate poor outcomes. For these wives, rather than standing by passively, they decided to use implants since they could be used covertly and would satisfy what they desired.

*"He said before that he would disagree, and if I did not listen, there would be a conflict. In the beginning, I was hesitant, but my need to use contraceptives was significant, so I decided to use a hidden type called the implant"*. (Muslim wife, monogamous, three children, Omdurman).

Other wives, especially those with advanced education and employment status, were strong in their views that contraceptive use is essentially and entirely a woman's decision. Disregarding cultural norms that may impose restrictions on women's contraceptive choices, these wives relied on their sound knowledge about the benefits of using family planning, determining which methods to use and when to use.

*"Family planning is important for everyone, and the wife decides when to use it according to her health. There is no such thing as a main decision maker (husband) or not. It was my choice to use the IUD method, then after my third child, I removed it to get pregnant again"*. (Muslim wife, monogamous, four children, Omdurman).

Wives also report that their engagement in advancing their professional positions and educational careers was essential to their lives. Whether they desire to continue or postpone their reproductive role, it is their choice and based on their decision.

"*If a woman wants to continue having children, it is her decision, and if she wants to use a contraceptive, it is her decision. I wanted to continue with my studies, so I decided to use the pill*". (Christian wife, monogamous, five children, Khartoum).

## Discussion

The findings of this study explore how husbands' preference for increased fertility is reflected in their wives' actual use of contraceptives and their wives ' responses to prevalent assumptions about gender dynamics in contraceptive decisions.

In considering the sociocultural interpretations of male discourse in much of Sub-Saharan Africa, depicting structures of unilateral decision-making [56], assuming more marital power [57], and expressions of male prestige by way of having large families [23], the study found that no matter the husband's socio-economic and educational status, they invariably wished for large families. A behaviour justified by upholding their socio-religious belief of increased procreation [30,58,59].

Similar findings from Kenya, Uganda [60], and Ghana [61] report the interrelation between a woman's use of family planning and the impact of gender inequity in contraceptive decisions. Women are expected to follow sociocultural constructs of increased childbearing behaviours, most likely agreeing with their husband's fertility choices and contraceptive decisions [62], as is evident in the findings of this study.

In contrast, the study findings reveal important outcomes where wives with different perceptions of fertility than their husbands and who were concerned about their reproductive health combined elements that could lead to increased use of family planning methods. Many of the wives believed that increased childbearing could create a burden on their health, encouraging them to continue with contraceptive use. This behaviour was supported by their increased awareness about different methods, more so than their husband, allowing them to make independent, informed decisions. The contraceptive pill was more favourable, as confirmed in previous studies [16,19].

Many wives took advantage of the present structures of the women-centeredness of family planning services, compounded by the cumulative experience of family planning providers toward women [10], enabling them to make decisions regarding their desired methods easily. Wives did report that husbands were not involved in their family planning visits. This was considered a women's space, and they were not inclined to share it. Although studies do posit that the social perception that men themselves consider family planning services a women's space and concern [63] and their exclusion from family planning could further contribute to decision discord among couples [64,34], it would, therefore, be more appropriate to initiate interventions suitable to improving men's involvement, without infringing on women's autonomy to family planning use. However, in this study, wives were more comfortable being present at family planning centres alone, which was also an opportunity to ensure privacy in discussions with health providers, particularly since discussions with husbands about their contraceptive needs led to cumbersome outcomes. This finding, therefore, implies that the context of the couple's discussion seemed atypical, prompting some of the wives to use long-acting methods covertly. Although for some wives, these may not be an attractive choice per se, they were encouraged to do so based on their candid need for contraceptives. The findings underscore the need for family planning programs to address the importance of couple counselling to facilitate women's desired choice of method [65,66].

The findings of this study further posit that wives' means to actualize their use of family planning methods were reinforced mainly by their advanced education and economic and urban backgrounds. For example, doctors' or engineers' wives were eminent in their perceptions that a woman's right to use their desired method and when to use it was their decision without being influenced or subjected to spousal interference. Findings from Egypt [67] and Ethiopia [37] reveal that not only does a woman's advanced socioeconomic status increase their decision-making power in contraceptive use,

but they are also more likely to challenge existing norms that perpetuate women's low use of contraceptive methods. Further, wives who were more likely to advance their careers were seen to delay their childbearing roles, increasing their demand for contraceptive use.

Whilst this study has contributed to the literature of social construction of gender roles and constructs of agency in contraceptive use and fertility-related behaviours among couples, this study has some limitations. This study examined only married couples' family planning decision-making dynamics. In this context, family planning services are offered mainly to married people. For future studies, a larger sample of men and women may be advisable to be carried out in other regions of Sudan to examine region-specific barriers or facilitators affecting women's contraceptive use. Notwithstanding, to our knowledge, this is the first study in Sudan to collect data using couple-level interviews by pairing husbands' and wives' responses to examine perspectives about fertility and family planning decisions, increasing the validity of findings.

### Study implications

The research findings posit the need to explore existing family planning initiatives and address the socio-cultural factors and gender inequity affecting women's contraceptive use. Initiatives in sub-Saharan Africa [68,69] proved to be successful in integrating men in family planning programs and harnessing their role in promoting women's contraceptive use [70,71]. It is suggested, therefore, that national reproductive health policies recognize the critical role health providers can play in engaging men. This requires investing in providers' skills to tailor their roles in counselling men about the health benefits of contraceptive use. In view of this study's context, where discussions of matters related to contraception and sexuality are considered a private issue among men [72], paying attention to the appropriate recruitment and training of health providers is noteworthy. Studies from Uganda [38] and Botswana [73] report that men would prefer same gender health providers. This was perceived to ensure feelings of comfort and trust among men in discussions concerning family planning, which could lead to increased involvement.

### Conclusion and recommendations

Partnerships between health officials at the policy level, researchers, and community health workers are needed to develop interventions encouraging men's involvement in family planning. Continuous research is recommended to examine the underlying elements of socio-cultural masculine norms to produce a more holistic view of why men are stereotyped as barriers to family planning use. Pilot data for such research could be collected through community interventions, which could serve as advocacy programs and/or public campaigns to increase awareness about family planning and reproductive health. Moreover, since the socio-religious values are closely intertwined with men's beliefs of increased fertility behaviours as has been evident in the findings of the study, it is also recommended to involve religious leaders who may play an influential role to engage in dialogues with husbands acknowledging that changes in perceptions of men's desire for increased fertility may not have to happen at the expense of the social image of manhood and virility. Rather, promoting their role as advocates and supporters of their wives' fertility health.

### Supporting information

**S1 Appendix. Interview questions for husband and wife participants.**
(DOCX)

### Acknowledgments

We want to express our gratitude to Ahfad University for Women, the medical staff of Fertility and Reproductive Health Services Centre (FRHC) in Khartoum, and Ahfad Family Health Centre (AFHC) in Omdurman. Thanks, and recognition also goes to this study's data collectors and respondents for sharing their time and personal views. All authors read and

approved the final manuscript. Professor Bart Van de Borne passed away before the submission of the final version of this manuscript. Dina Badri accepts responsibility for the integrity and validity of the data collected and analyzed.

## Author contributions

**Conceptualization:** Dina Badri.

**Data curation:** Dina Badri.

**Formal analysis:** Dina Badri.

**Investigation:** Dina Badri.

**Methodology:** Dina Badri, Anja Krumeich, Bart Van de Borne.

**Project administration:** Dina Badri, Anja Krumeich, Bart Van de Borne.

**Resources:** Dina Badri, Anja Krumeich, Bart Van de Borne.

**Supervision:** Dina Badri, Anja Krumeich, Bart Van de Borne.

**Validation:** Dina Badri, Anja Krumeich, Bart Van de Borne.

**Writing – original draft:** Dina Badri.

**Writing – review & editing:** Dina Badri, Anja Krumeich, Bart Van de Borne.

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
