## [Decision Letter · Decision Letter 0]

17 Oct 2024

Dear Dr. Badri,

Thank you for submitting your manuscript to PLOS ONE. After careful consideration, we feel that it has merit but does not fully meet PLOS ONE’s publication criteria as it currently stands. Therefore, we invite you to submit a revised version of the manuscript that addresses the points raised during the review process.

**We would like to be able to reconsider the manuscript and hope you can successfully address the concerns outlined below by the reviewers.**

**Reviewer #1**

Lack of Novelty:

The study contributes little new knowledge to the existing literature. The influence of male dominance in family planning decisions, particularly in patriarchal societies, is well-documented in various studies across sub-Saharan Africa. This manuscript does not offer new insights or significant theoretical advancements in understanding these dynamics.

Methodological Flaws:

The qualitative methodology employed is insufficiently rigorous. The study’s sample size of 46 participants (23 couples) is small and lacks diversity, limiting the generalizability of the findings. Additionally, the recruitment process using a snowball sampling method introduces significant bias, as participants are likely to share similar sociocultural backgrounds, further reducing the representativeness of the sample.

The data collection and analysis methods are described in vague terms, with insufficient detail about how thematic coding was conducted. The use of NVivo software is mentioned, but without a clear explanation of the coding process, it is difficult to assess the reliability and validity of the results.

Overemphasis on Patriarchy without Critical Analysis:

The manuscript heavily emphasizes the role of patriarchy in shaping family planning decisions but fails to critically analyze this context. The discussion lacks depth and does not engage with alternative explanations or theoretical frameworks that could provide a more nuanced understanding of the findings. For instance, the role of economic factors, education levels, or healthcare access is not adequately explored.

Repetitiveness and Redundancy:

The manuscript is repetitive, particularly in the results and discussion sections. The same points about male dominance and decision-making power are reiterated multiple times without adding substantial new information or analysis. This repetition detracts from the manuscript's overall clarity and coherence.

Limited Practical Implications:

While the study aims to inform policymakers about the importance of male involvement in family planning, it does not provide actionable recommendations or innovative strategies to address the identified issues. The suggestions made are generic and lack specificity, reducing the potential impact of the research on policy and practice.

Ethical Considerations and Reflexivity:

The manuscript does not adequately address ethical considerations or the potential impact of the research on participants. Given the sensitive nature of the topic, more attention should have been given to ensuring confidentiality, especially considering the patriarchal context where discussing family planning can have serious social implications. Additionally, there is a lack of reflexivity regarding the researchers' positions and how this might have influenced the study.

Specific Comments:

Introduction:

The introduction provides a broad overview of the issue but lacks a clear statement of the research question. It would benefit from a more focused articulation of the study’s objectives and how it aims to fill specific gaps in the existing literature.

Research Methods:

The description of the qualitative approach is inadequate. The manuscript should provide more detail on the interview process, including how questions were structured and how interviews were conducted. The justification for the chosen sample size and the use of snowball sampling is weak, and these limitations should be acknowledged more transparently.

Results:

The results section is overly descriptive and lacks critical analysis. The themes identified from the interviews are not sufficiently supported by direct quotes or examples from the data, making it difficult to assess the validity of the interpretations.

Discussion:

The discussion fails to engage critically with the findings. It merely restates the results without offering deeper insights or considering alternative interpretations. The manuscript would benefit from a more critical engagement with the literature and a discussion of the broader implications of the findings.

Conclusion:

The conclusion does not effectively summarize the contributions of the study or suggest clear directions for future research. It should offer more specific recommendations for policy and practice, grounded in the study’s findings.

Recommendation:

Given the significant methodological weaknesses, lack of novelty, and inadequate analysis, I recommend rejection of the manuscript in its current form. The study does not make a substantial contribution to the field, and the identified flaws undermine the credibility and relevance of the findings.

**Reviewer # 2**

TITTLE

Improve the title by unnecessary word.Title to be reversed

ABSTRACT

METHODS

What is your study design? I comment mention the study design then approachYou interview the husband or their wives make it clear

CONCLUSION

Not well addressConclude based on the key findings

INTRODUCTION

NOT well addressed with your title be specific. I comment to be reversed.Improve on writing much citation little voice of those researchersShould explained the role of husband and how will influence family planningYou explain little modern family planning but natural family planning not well explainAnd how about 9% is prevalence of modern or natural be specific and improve

METHODS

How do you perform the interview?What are the roles of researcher data collector?Use the researcher scientific wordsIt’s thematic or content analysis makes it clearWhat is the procedure done?How trustworthiness was ensured?Ethical consideration should be well described to PLOS ONE guideline?

RESULT

On part of result I comment to include the social demographic table.Also I comment using term majority on your result.I comment writing the number of respondent when you make the quotation.Also the tittle does not relate with the result especially in the quotations there is no husband role decision making explained its seem like you explain more perceptions and factors of decision making due to their wives. Make it reverse or change your title to write perceptionsIn the part of questions are too long and appeared to be like sentences. Improve the quotations

REFFERENCE

Should be reversed are not clear in PLOS ONE guideline

DISCUSSION

Should be reversed.I comment relate with the result

We look forward to receiving your revised manuscript.

Kind regards,

Godwin Banafo Akrong, Ph.D.

Academic Editor

PLOS ONE

**Journal Requirements:**

2. We note that this data set consists of interview transcripts. Can you please confirm that all participants gave consent for interview transcript to be published?

If they DID provide consent for these transcripts to be published, please also confirm that the transcripts do not contain any potentially identifying information (or let us know if the participants consented to having their personal details published and made publicly available). We consider the following details to be identifying information:

- Names, nicknames, and initials

- Age more specific than round numbers

- GPS coordinates, physical addresses, IP addresses, email addresses

- Information in small sample sizes (e.g. 40 students from X class in X year at X university)

- Specific dates (e.g. visit dates, interview dates)

- ID numbers

Or, if the participants DID NOT provide consent for these transcripts to be published:

- Provide a de-identified version of the data or excerpts of interview responses

- Provide information regarding how these transcripts can be accessed by researchers who meet the criteria for access to confidential data, including:

a) the grounds for restriction

b) the name of the ethics committee, Institutional Review Board, or third-party organization that is imposing sharing restrictions on the data

c) a non-author, institutional point of contact that is able to field data access queries, in the interest of maintaining long-term data accessibility.

d) Any relevant data set names, URLs, DOIs, etc. that an independent researcher would need in order to request your minimal data set.

For further information on sharing data that contains sensitive participant information, please see: https://journals.plos.org/plosone/s/data-availability#loc-human-research-participant-data-and-other-sensitive-data

If there are ethical, legal, or third-party restrictions upon your dataset, you must provide all of the following details (https://journals.plos.org/plosone/s/data-availability#loc-acceptable-data-access-restrictions):

a) A complete description of the dataset

b) The nature of the restrictions upon the data (ethical, legal, or owned by a third party) and the reasoning behind them

c) The full name of the body imposing the restrictions upon your dataset (ethics committee, institution, data access committee, etc)

d) If the data are owned by a third party, confirmation of whether the authors received any special privileges in accessing the data that other researchers would not have

e) Direct, non-author contact information (preferably email) for the body imposing the restrictions upon the data, to which data access requests can be sent.

Reviewers' comments:

Reviewer's Responses to Questions

**Comments to the Author**

1. Is the manuscript technically sound, and do the data support the conclusions?

Reviewer #1: No

Reviewer #2: Yes

2. Has the statistical analysis been performed appropriately and rigorously?

Reviewer #1: No

Reviewer #2: Yes

3. Have the authors made all data underlying the findings in their manuscript fully available?

Reviewer #1: Yes

Reviewer #2: Yes

4. Is the manuscript presented in an intelligible fashion and written in standard English?

Reviewer #1: Yes

Reviewer #2: Yes

**Reviewer #1:** Overall Summary:

The manuscript explores the role of Sudanese husbands in the decision-making process related to their wives' family planning use in Khartoum and Omdurman. The study is grounded in qualitative research methods, involving interviews with 46 participants (23 married couples) to understand the sociocultural and gender norms influencing family planning decisions.

Major Concerns and Weaknesses:

Lack of Novelty:

The study contributes little new knowledge to the existing literature. The influence of male dominance in family planning decisions, particularly in patriarchal societies, is well-documented in various studies across sub-Saharan Africa. This manuscript does not offer new insights or significant theoretical advancements in understanding these dynamics.

Methodological Flaws:

The qualitative methodology employed is insufficiently rigorous. The study’s sample size of 46 participants (23 couples) is small and lacks diversity, limiting the generalizability of the findings. Additionally, the recruitment process using a snowball sampling method introduces significant bias, as participants are likely to share similar sociocultural backgrounds, further reducing the representativeness of the sample.

The data collection and analysis methods are described in vague terms, with insufficient detail about how thematic coding was conducted. The use of NVivo software is mentioned, but without a clear explanation of the coding process, it is difficult to assess the reliability and validity of the results.

Overemphasis on Patriarchy without Critical Analysis:

The manuscript heavily emphasizes the role of patriarchy in shaping family planning decisions but fails to critically analyze this context. The discussion lacks depth and does not engage with alternative explanations or theoretical frameworks that could provide a more nuanced understanding of the findings. For instance, the role of economic factors, education levels, or healthcare access is not adequately explored.

Repetitiveness and Redundancy:

The manuscript is repetitive, particularly in the results and discussion sections. The same points about male dominance and decision-making power are reiterated multiple times without adding substantial new information or analysis. This repetition detracts from the manuscript's overall clarity and coherence.

Limited Practical Implications:

While the study aims to inform policymakers about the importance of male involvement in family planning, it does not provide actionable recommendations or innovative strategies to address the identified issues. The suggestions made are generic and lack specificity, reducing the potential impact of the research on policy and practice.

Ethical Considerations and Reflexivity:

The manuscript does not adequately address ethical considerations or the potential impact of the research on participants. Given the sensitive nature of the topic, more attention should have been given to ensuring confidentiality, especially considering the patriarchal context where discussing family planning can have serious social implications. Additionally, there is a lack of reflexivity regarding the researchers' positions and how this might have influenced the study.

Specific Comments:

Introduction:

The introduction provides a broad overview of the issue but lacks a clear statement of the research question. It would benefit from a more focused articulation of the study’s objectives and how it aims to fill specific gaps in the existing literature.

Research Methods:

The description of the qualitative approach is inadequate. The manuscript should provide more detail on the interview process, including how questions were structured and how interviews were conducted. The justification for the chosen sample size and the use of snowball sampling is weak, and these limitations should be acknowledged more transparently.

Results:

The results section is overly descriptive and lacks critical analysis. The themes identified from the interviews are not sufficiently supported by direct quotes or examples from the data, making it difficult to assess the validity of the interpretations.

Discussion:

The discussion fails to engage critically with the findings. It merely restates the results without offering deeper insights or considering alternative interpretations. The manuscript would benefit from a more critical engagement with the literature and a discussion of the broader implications of the findings.

Conclusion:

The conclusion does not effectively summarize the contributions of the study or suggest clear directions for future research. It should offer more specific recommendations for policy and practice, grounded in the study’s findings.

Recommendation:

Given the significant methodological weaknesses, lack of novelty, and inadequate analysis, I recommend rejection of the manuscript in its current form. The study does not make a substantial contribution to the field, and the identified flaws undermine the credibility and relevance of the findings.

**Reviewer #2:** Generally the study appeared to be very important as it addressed the impact of quality of health care services and well-being of maternal and perinatal good outcome .HOWEVER many issues need to be addressed to make it more clear and scientific

Follow to the PLOS ONE in organizing your work

TITTLE

Improve the title by unnecessary word.

Title to be reversed

ABSTRACT

METHODS

What is your study design? I comment mention the study design then approach

You interview the husband or their wives make it clear

CONCLUSION

Not well address

Conclude based on the key findings

INTRODUCTION

NOT well addressed with your title be specific. I comment to be reversed.

Improve on writing much citation little voice of those researchers

Should explained the role of husband and how will influence family planning

You explain little modern family planning but natural family planning not well explain

And how about 9% is prevalence of modern or natural be specific and improve

METHODS

How do you perform the interview?

What are the roles of researcher data collector?

Use the researcher scientific words

It’s thematic or content analysis makes it clear

What is the procedure done?

How trustworthiness was ensured?

Ethical consideration should be well described to PLOS ONE guideline?

RESULT

On part of result I comment to include the social demographic table.

Also I comment using term majority on your result.

I comment writing the number of respondent when you make the quotation.

Also the tittle does not relate with the result especially in the quotations there is no husband role decision making explained its seem like you explain more perceptions and factors of decision making due to their wives. Make it reverse or change your title to write perceptions

In the part of questions are too long and appeared to be like sentences. Improve the quotations

REFFERENCE

Should be reversed are not clear in PLOS ONE guideline

DISCUSSION

Should be reversed.

I comment relate with the result

N:B;

This is a very interesting paper, which is scientifically ground; the authors should work on those comments to improve the paper.

Thank You.

**Do you want your identity to be public for this peer review?** For information about this choice, including consent withdrawal, please see our Privacy Policy

Reviewer #1: No

Reviewer #2: No

---

## [Author Response · Author response to Decision Letter 1]

24 Dec 2024

To the Academic Editor PLoS One

Subject: Re-submitting the revised manuscript. Title: Decisions and Choices about Fertility and Family Planning: Perspectives from Husbands and Wives in Sudan.

Thank you for availing me the opportunity to re-submit the revised manuscript.

As my ongoing PhD thesis, this manuscript is part of my larger study, which examines interrelated aspects of fertility and family planning.

This study was conceptualised with the understanding that in Sudan, married women’s low access to family planning has a detrimental impact on their reproductive health or results in unplanned pregnancies.

In previous studies in Sudan, the low contraceptive use of married women was associated with factors concerning their low socio-economic status, misconceptions of the use of contraceptives, and other cultural practices leading to problems in accessing and using family planning services.

While these studies provide valuable evidence about family planning use, The researcher's position and previous professional experience in reproductive health have allowed her to recognize that discussions about women alone, like discussions about men separately, is an insufficient approach to understanding fertility and family planning practices. In Sudan, men, in general, are notably understudied in fertility deliberations and the natural realm of family planning literature. Their influence and role as supportive or unsupportive to their wives in the Sudanese context needs to be examined.

Furthermore, in Sub Saharan African context and elsewhere, particularly where fertility remains high, a sizeable mass of studies has claimed the importance of scaling up family planning programs to examine gender norms whilst recognizing men’s role in their partner's contraceptive use. However, what is missing from the literature is to what extent are men’s behaviours and attitudes influencing a women’s fertility and family planning use.

This study, therefore, aims to present a more nuanced understanding of whether husbands' fertility preferences are actually reflected in their wives' family planning practices and how and when their wives adopt their wishes in their actual family planning use.

As mentioned in my previous letter dated March 2024, given the current ongoing war in Sudan, we hope that in the not-so-far future, Sudan will get back on track and the significance of this paper will contribute to realising underlying factors that relate to the country’s low contraceptive prevalence rate.

In this regard, I would like to put forth the following comments.

What is known about this topic

• Ample scientific evidence from sub-Saharan Africa and elsewhere has reported the role of men’s decision-making role in family planning.

• Men’s disapproval and lack of engagement in family planning contribute to women’s low levels of contraceptive use.

• There are cultural traditions perceived to influence women’s use of family planning in contexts with high fertility.

What this study adds

• This is one of few studies in a sub-Saharan African setting and the first in Sudan to match the husband and wife's responses in fertility and family planning literature.

• The study provides information about the socio-cultural influences that shape divergent perceptions of fertility preferences and family planning use of women.

• The study presents findings based on the context and extent to which wives are willing to concede or overrule men’s preferences in fertility and family planning.

Response to Academic editor, reviewers 1 and 2 are noted below. Please note that there are episodes of overlap in responses to the reviewers, but that is because there are some similarities in points put forward by them.

Academic Editor:

Please see below.

We note that this data set consists of interview transcripts. Can you please confirm that all participants gave consent for interview transcript to be published?

If they DID provide consent for these transcripts to be published, please also confirm that the transcripts do not contain any potentially identifying information (or let us know if the participants consented to having their personal details published and made publicly available). We consider the following details to be identifying information:

- Names, nicknames, and initials

- Age more specific than round numbers

- GPS coordinates, physical addresses, IP addresses, email addresses

- Information in small sample sizes (e.g. 40 students from X class in X year at X university)

- Specific dates (e.g. visit dates, interview dates)

- ID numbers

In The methods section of the study it has been mentioned that the participants have been informed of the study's nature and purpose. Accordingly, all eligible participants agreed to participate and were assured of anonymity. Confirming this point, as shown in the results section of the study, quotations from the study participants were used to reflect and respond to the study’s main aim. captions/quotes. The names, specific ages, addresses, etc., are not shown in the study.

Please amend either the title on the online submission form (via Edit Submission) or the title in the manuscript so that they are identical.

The title has been amended as per comments from reviewer 2.

Please include captions for your Supporting Information files at the end of your manuscript, and update any in-text citations to match accordingly. As mentioned in the methods section, Annex 1 shows questions that guided the interviews with participants. Tables 1 and 2, included in the study, show the socio-demographic background of the participants and the use of the couple’s family planning methods, respectively. In-text citations/quotes do match according to socio-demographic characteristics, as shown in Table 1.

No changes

We look forward to receiving your revised manuscript.

I look forward to a fruitful collaboration and a beneficial contribution to the family planning literature about Sudan.

Reviewer 1:

Comment:

Lack of Novelty:

The study contributes little new knowledge to the existing literature. The influence of male dominance in family planning decisions, particularly in patriarchal societies, is well-documented in various studies across sub-Saharan Africa. This manuscript does not offer new insights or significant theoretical advancements in understanding these dynamics.

As some studies indeed show men’s decision-making role in women’s family planning, The authors of this study have critically reviewed it to contribute to the literature of family planning and reproductive health to address specifically to what extent are men’s fertility perceptions and attitudes actually influencing and reflected in their wives' contraceptive use and by explaining how their wives' react to their husbands' influences.

Specific Comments:

Introduction:

The introduction provides a broad overview of the issue but lacks a clear statement of the research question. It would benefit from a more focused articulation of the study’s objectives and how it aims to fill specific gaps in the existing literature.

The introduction has been revised to posit Sudan’s high fertility contexts and the low contraceptive prevalence rate and what factors might contribute to this. The study also highlights the men’s fertility desires as one of the reasons behind a women’s low contraceptive use, which is relevant to the context of Sudan. The introduction ends with the main aim of the study, as indicated above. Moreover, studies from Sub-Saharan Africa and elsewhere do present men’s decision-making behaviors; however, to what extent these behaviors influence women are still under studied. Some agree to their decisions, yet other women may not for several reasons, which may indicate a shift in power dynamics among married couples. Furthermore, the examination of responses from husbands and their wives presents a more nuanced understanding in this regard.

Methodological Flaws:

The qualitative methodology employed is insufficiently rigorous. The study’s sample size of 46 participants (23 couples) is small and lacks diversity, limiting the generalizability of the findings. Additionally, the recruitment process using a snowball sampling method introduces significant bias, as participants are likely to share similar sociocultural backgrounds, further reducing the representativeness of the sample.

The data collection and analysis methods are described in vague terms, with insufficient detail about how thematic coding was conducted. The use of NVivo software is mentioned, but without a clear explanation of the coding process, it is difficult to assess the reliability and validity of the results.

Regarding the scope of this study's participants, inclusion was continued until saturation was observed, and accordingly, 46 participants (23 Couples) representing different socio-economic, religious, and educational backgrounds and family sizes were included.

The methodology has been revised, and a detailed description using subheadings identifies the procedure and validity of the study results.

Research Methods:

The description of the qualitative approach is inadequate. The manuscript should provide more detail on the interview process, including how questions were structured and how interviews were conducted. The justification for the chosen sample size and the use of snowball sampling is weak, and these limitations should be acknowledged more transparently.

More detail about interviewing, selection of participants purposively from the catchment areas of the family planning centers in Khartoum and Omdurman areas and how interviews with the married couple has been detailed in the methods section.

In the limitation section, it is mentioned that a larger sample for future research in a similar study is recommended. However, using a qualitative approach allows for more in-depth examination of perceptions and behaviours responding to questions of how and when husbands and wives perceive the use of family planning.

Results:

The results section is overly descriptive and lacks critical analysis. The themes identified from the interviews are not sufficiently supported by direct quotes or examples from the data, making it difficult to assess the validity of the interpretations.

The results section has been revised and amended based on data collected from the participants. The use of quotes in the study allows the reader to engage more closely with the perception and behavior of the participants. For example, in the interviews, some of the husbands referred to local proverbs, which are an explanation of how men in the Sudanese context view and justify their positionality when it comes to making decisions. this provides an opportunity for the reader to learn about the underlying reasoning for masculine behaviors. However, this does not engulf/encompass the whole of the study, recognizing comments of reviewers point about heavily emphasising patriarchs. It only lays the ground to contextualize how wives, despite recognizing patriarchs, may overrule such dominance.

Overemphasis on Patriarchy without Critical Analysis:

The manuscript heavily emphasizes the role of patriarchy in shaping family planning decisions but fails to critically analyze this context. The discussion lacks depth and does not engage with alternative explanations or theoretical frameworks that could provide a more nuanced understanding of the findings. For instance, the role of economic factors, education levels, or healthcare access is not adequately explored.

The majority of the husbands shared the concept of patriarch in this study regardless of their economic and educational characteristics, as indicated in the results section, largely based on their socialised upbringing. However, a wife’s response to these notions of masculinity may not be applicable to all of them, particularly in view of their reproductive health and contractive needs.

Discussion:

The discussion fails to engage critically with the findings. It merely restates the results without offering deeper insights or considering alternative interpretations. The manuscript would benefit from a more critical engagement with the literature and a discussion of the broader implications of the findings.

The discussion has been reviewed and revised to engage with the findings,

Repetitiveness and Redundancy:

The manuscript is repetitive, particularly in the results and discussion sections. The same points about male dominance and decision-making power are reiterated multiple times without adding substantial new information or analysis. This repetition detracts from the manuscript's overall clarity and coherence.

The results have been revised to highlight the emerging themes, which cover specifically the study's argument: agreement over fertility and family planning, disagreement about fertility and family planning, and the change of power deliberated by the wives in addressing their contraceptive needs in spite of being aware of notions of husbands' influence. They have done so in different ways to safeguard their reproductive health.

Limited Practical Implications:

While the study aims to inform policymakers about the importance of male involvement in family planning, it does not provide actionable recommendations or innovative strategies to address the identified issues. The suggestions made are generic and lack specificity, reducing the potential impact of the research on policy and practice.

The conclusion has been revised to posit the future role of family planning providers and relevant stakeholders to recognize that gender approach could be integrated into future family planning programs is essential for its success and that a women’s empowerment and visibility to attain her contraceptive choice can be achieved then. Also, men’s family planning literacy is an important step forward to recognise the importance of family planning use and its health benefits for women.

Ethical Considerations and Reflexivity:

The manuscript does not adequately address ethical considerations or the potential impact of the research on participants. Given the sensitive nature of the topic, more attention should have been given to ensuring confidentiality, especially considering the patriarchal context where discussing family planning can have serious social implications. Additionally, there is a lack of reflexivity regarding the researchers' positions and how this might have influenced the study.

Given the sensitive nature of this study with particular reference to husbands, participants were informed about study objectives, and after obtaining their verbal consent and agreeing to participate, the interviews were conducted by male and female data collectors, each corresponding to the same gender. Interviews were also conducted at the discretion of the participants in terms of time and place. However, always in separate spaces so that responses are not heard or influenced by either of the couple.

Conclusion:

The conclusion does not effectively summarize the contributions of the study or suggest clear directions for future research. It should offer more specific recommendations for policy and practice, grounded in the study’s findings.

This has been amended to summarise what the study recommends for future family planning programs for the Sudan context, which can also be relevant to other similar contexts of high fertility women’s low contraceptive use and lack of men's involvement.

Reviewer 2:

Comment:

TITTLE

• Improve the title by unnecessary word.

• Title to be reversed the manuscript title and keywords have been changed

ABSTRACT

This section has been revised to reflect a summary of the amended study.

INTRODUCTION

• NOT well addressed with your title be specific. I comment to be reversed.

• Improve on writing much citation little voice of those researchers

• Should explained the role of husband and

---

## [Decision Letter · Decision Letter 1]

29 Jan 2025

Dear Dr. Badri,

Thank you for submitting your manuscript to PLOS ONE. After careful consideration, we feel that it has merit but does not fully meet PLOS ONE’s publication criteria as it currently stands. Therefore, we invite you to submit a revised version of the manuscript that addresses the points raised during the review process.

**ACADEMIC EDITOR:**

Reviewer #2 requires you to add a table showing the social demographics of your respondents.Try to indicate the study design and approach clearly. (Reviewer #2)How did you ensure trustworthiness in your study? (Reviewer #2)Indicate clearly whether you used thematic or content analysis. (Reviewer #2)Kindly revise and improve upon your limitations. (Reviewer #2)Kindly make sure to proofread your work and correct all grammatical and typographical errors. (Reviewer #2)Introduction: This section needs revision. There are multiple short paragraphs, they lack continuity and do not establish facts in a coherent manner. (Reviewer #3)Results: The quantitative information about respondents should be given as summary tables and not for each respondent. (Reviewer #3)The qualitative results need further efforts by the authors. They are given in a very superfluous manner and lack an in-depth analysis. The themes and sub-themes emerging from the data need to be described in more detail. (Reviewer #3)Discussion: I cannot comment on this section as the Results are not described comprehensively. (Reviewer #3)

We look forward to receiving your revised manuscript.

Kind regards,

Godwin Banafo Akrong, Ph.D.

Academic Editor

PLOS ONE

Journal Requirements:

Reviewers' comments:

Reviewer's Responses to Questions

**Comments to the Author**

Reviewer #2: All comments have been addressed

Reviewer #3: (No Response)

2. Is the manuscript technically sound, and do the data support the conclusions?

Reviewer #2: Yes

Reviewer #3: Partly

3. Has the statistical analysis been performed appropriately and rigorously?

Reviewer #2: Yes

Reviewer #3: No

4. Have the authors made all data underlying the findings in their manuscript fully available?

Reviewer #2: Yes

Reviewer #3: Yes

5. Is the manuscript presented in an intelligible fashion and written in standard English?

Reviewer #2: Yes

Reviewer #3: No

Reviewer #2: REVIEWER COMMENT AND SUGGESTIONS

Generally congratulation to the authors on putting much effort on improving this work however few issues should make it clear to the reader.

• Work extensively to be clear grammar and typographical errors throughout the document

• Also on top of your title authors should write the study design

• On part of result I comment to include the social demographic table it will be more better.

• What is your study design? I comment mention the study design then approach

• How trustworthiness was ensured

• It’s thematic or content analysis makes it clear

• On part of limitation the authors should revise and improve

NB:

Congratulation again to the authors the manuscript now It have improvement.

Reviewer #3: The research work addresses an important issue however, it needs to be written in a better manner for publication.

Introduction: This section needs revision. There are multiple short paragraphs, they lack continuity and do not establish facts in a coherent manner.

Results: The quantitative information about respondents should be given as summary tables and not for each respondent. The qualitative results need further efforts by the authors. They are given in a very superfluous manner and lack an in-depth analysis. The themes and sub-themes emerging from the data need to be described in more detail.

Discussion: I can not comment on this section as the Results are not described comprehensively.

**Do you want your identity to be public for this peer review?** For information about this choice, including consent withdrawal, please see our Privacy Policy

Reviewer #2: **Yes:** rehema abdallah

Reviewer #3: No

---

## [Author Response · Author response to Decision Letter 2]

4 Mar 2025

Academic editor comments:

Please review your reference list. Changes to the reference list should be mentioned in the rebuttal letter

Revisions have been made accordingly within the text and the reference list.

Reviewer 2 comments:

Add a table showing the social demographics of your respondents. Table has been included, please refer to the results section.

Indicate the study design and approach clearly. Methodology has been revised to clarify design and approach.

How did you ensure trustworthiness in your study. Methodology has been revised to clarify the trustworthiness of study.

Indicate clearly whether you used thematic or content analysis. Methodology has been revised to clarify this.

Kindly revise and improve upon your limitations. Limitation revised.

Kindly make sure to proofread your work and correct all grammatical and typographical errors. Re-read and corrected.

Reviewer 3:

Introduction: This section needs revision. There are multiple short paragraphs, they lack continuity and do not establish facts in a coherent manner. The introduction has been changed and structured to reflect continuity and coherence.

The quantitative information about respondents should be given as summary tables and not for each respondent.

S1 table presenting socio-demographic details and S2 table presenting the number of women using family planning has been summarised.

The qualitative results need further efforts by the authors. They are given in a very superfluous manner and lack an in-depth analysis. The themes and sub-themes emerging from the data need to be described in more detail. You explain little modern family planning but natural family planning not well explained.

The results section has been re-written. The use of participants’ narratives and the emerging analysis from the data presents a clearer results section.

I cannot comment on this section as the Results are not described comprehensively. The discussion section has been re-written to present a summary of findings in a concise manner.

---

## [Decision Letter · Decision Letter 2]

27 Mar 2025

Dear Dr. Badri,

Thank you for submitting your manuscript to PLOS ONE. After careful consideration, we feel that it has merit but does not fully meet PLOS ONE’s publication criteria as it currently stands. Therefore, we invite you to submit a revised version of the manuscript that addresses the points raised during the review process.

Key Areas for Improvement:

Kindly address the concerns raised regarding the research methods, results, and discussions.

In summary, I encourage you to address all the reviewers' comments and make the necessary revisions. I look forward to reviewing your revised manuscript.

We look forward to receiving your revised manuscript.

Kind regards,

Godwin Banafo Akrong, Ph.D.

Academic Editor

PLOS ONE

Journal Requirements:

Reviewers' comments:

Reviewer's Responses to Questions

**Comments to the Author**

Reviewer #2: All comments have been addressed

Reviewer #3: (No Response)

2. Is the manuscript technically sound, and do the data support the conclusions?

Reviewer #2: Yes

Reviewer #3: Partly

3. Has the statistical analysis been performed appropriately and rigorously?

Reviewer #2: Yes

Reviewer #3: N/A

4. Have the authors made all data underlying the findings in their manuscript fully available?

Reviewer #2: Yes

Reviewer #3: Yes

5. Is the manuscript presented in an intelligible fashion and written in standard English?

Reviewer #2: Yes

Reviewer #3: Yes

Reviewer #2: Reviewer Comment and suggestions

Abstract:

• The abstract is well-structured; however, consider summarizing the key findings in bold points for easier readability.

Introduction:

• Clearly define key terms early on, such as "family planning," "contraceptive use," and specific cultural terms related to fertility in Sudan.

• The authors could strengthen the rationale for the study by explicitly stating the gaps in existing research and the significance of addressing these in the Sudanese context.

Methodology:

• The authors should have provided strong details on methodology; however, clarify the rationale behind the sample size (46) and the decision to conduct interviews separately.

• Consider including additional information on any pilot testing done for interview guides or data collection methods to enhance credibility.

Results

• The four pathways are insightful. To enhance clarity, consider summarizing each pathway with a diagram or table that encapsulates the findings visually.

• When quoting participants, ensure a uniform presentation style to enhance readability.

• Tables (S1 and S2): Ensure that the tables are referenced consistently in the text. For instance, include contextual details before referencing a particular table.

Discussion

• Integrate relevant theoretical frameworks that support your findings. This could enhance the academic level and context of your analysis.

• Connect the findings to broader implications beyond Sudan, discussing how culturally influential decisions in family planning are a common issue in various regions globally.

Limitations:

• Your acknowledgment of limitations is good. It could be beneficial to offer suggestions for future research, highlighting how studies could be expanded or improved in method.

Conclusion

• The conclusion consolidates findings effectively. Reinforce the call to action for integrating men into family planning discussions, as it positions your research within a solution-focused frame.

NB

• While the tone is mostly formal and academic, consider refining some sentences for conciseness and impact.

• Ensure that all references are formatted consistently according to the selected citation style; this includes consistency in formatting year, volume, and pages in your references list.

• A thorough proofreading process could help catch minor typographical or grammatical errors that may have slipped through.

Reviewer #3: Research Methods:

The description of the qualitative approach is inadequate. The summary tables for the socio-demographic are not in the standard way. The justification for the chosen sample size and the use of snowball sampling is weak, and these limitations should be acknowledged more transparently.

Results:

The results section is still descriptive and lacks critical analysis. The themes identified from the interviews are supported by direct quotes or examples from the data, but the analytic approach for the results reported is weak.

Discussion:

The discussion does not critically engage with the findings. It merely restates the results without offering deeper insights or considering alternative interpretations. The manuscript would benefit from a more critical engagement with the literature and a discussion of the broader implications of the findings.

**Do you want your identity to be public for this peer review?** For information about this choice, including consent withdrawal, please see our Privacy Policy

Reviewer #2: **Yes:** rehema abdallah

Reviewer #3: No

---

## [Author Response · Author response to Decision Letter 3]

23 May 2025

Please find a response to the Academic editor, reviewers 2 and 3 noted below.

Academic Editor:

Please review your reference list. Changes to the reference list should be mentioned in the rebuttal letter

The text and the reference list have been revised accordingly. The additions and omitted references below are better suited to the revised manuscript.

Added references:

Caldwell JC, Caldwell P. The Cultural Context of High Fertility in Sub-Saharan Africa. Population and Development Review. 1987; 13 (3): 409–437. doi:10.2307/1973133.

Musalia J. Extra conjugal determinants of spousal communication about family planning in Kenya. Sex Roles. 2003; 49 (11): 597-607. doi: 10.1023/B:SERS.0000003130.04774.a1

Upadhyay UD, Karasek D. Women’s empowerment and achievement of desired fertility in Sub-Saharan Africa. DHS Working Papers 80. 2010; Calverton, MD: ICF Macro. 2010.

Bankole A, Ezeh AC. Unmet Need for Couples: An Analytical Framework and Evaluation with DHS Data. Population Research and Policy Review. 1999. 18 (6): 579-605. https://doi.org/10.1023/A:1006373106870

Kodzi I, Johnson D. Casterline JB. Examining the Predictive Value of Fertility Preferences Among Ghanaian Women. Demographic Research 22. 2010; (30): 965-984. doi:10.4054/DemRes.2010.22.30.

Randall S, Mondain N, Diagne A. Men, Fertility Control and Contraception in Senegal. Paper presented at Sixth African Population Conference, Ouagadougou. INDEPTH network, ACCRA Ghana, December 2011.

Anbesu EW, Aychiluhm SB, Alemayehu M. Women’s decisions regarding family planning use and its determinants in Ethiopia: A systematic review and meta-analysis protocol. PLoS ONE. 2022; 17(10): e0276128. https://doi.org/10.1371/journal.pone.0276128

Ahmed HM. Barriers to family planning in Sudan: Results from a survey in White Nile, Kassala and Al‐Gadarif, 2008. African Development Review. 2013; 25 (4): 499–512.

Patton MQ. Qualitative Evaluation and Research Methods (2nd ed.). Newbury Park, CA: Sage. 1990.

Nyimbili F. Nyimbili L. Types of purposive sampling techniques with their examples and application in qualitative research studies. British Journal of Multidisciplinary and Advanced Studies: English Lang., Teaching, Literature, Linguistics & Communication. 2024; 5(1),90-99

Bankole, A. and Singh, S. ‘Couples’ fertility and contraceptive decision-making in developing countries: Hearing the man’s voice’. International Family Planning Perspectives. 1998; 24:2. DOI: 10.2307/2991915

Alnory, AH. Islam, Women and Fertility in Sudan. Gezira Journal of Economic and Social Sciences 2. 2010; University of Gezira, Medani – SUDAN.

Blacker J, Opiyo C, Jasseh M, Sloggett A, Ssekamatte-Ssebuliba J. Fertility in Kenya and Uganda: A comparative study of trends and determinants. Popul Stud. 2005; 59(3):355–373.

Bawah AA, Akweongo P, Simmons R, Phillips JF. Women’s fears and men’s anxieties: The impact of family planning on gender relations in northern Ghana. Studies in Family Planning. 1999; 30(1): 54–66.

Agarwal B. Bargaining and gender relations: Within and beyond the household. Feminist Economics. 1997; 3:1:1–51. doi.org/10.1080/135457097338799

Wambete SN, Serwaa D, Dzantor EK, Baru A, Poku-Agyemang E, Kukeba MW, et al. Determinants for male involvement in family planning and contraception in Nakawa Division, Kampala, Uganda; An urban slum qualitative study. PLOS Glob Public Health. 2024; 4(5): e0003207. https://doi.org/10.1371/journal.pgph.0003207

Johnbosco M, Love O, Chuma E, Christian M, Chukwunenye I. Ifeoma E. Male involvement in family planning; An often neglected determinant of contraceptive prevalence in Sub-Saharan Africa. International Journal of Scientific Reports. 2019; 260–265; DOI:https://doi.org/10.18203/issn.2454-2156.IntJSciRep20193766.

Biddlecom AE, Fapohunda BM. Covert contraceptive use: Prevalence, motivations, and consequences. Stud Fam Plan. 1998;29(4):360–72.

Kibira SPS, Karp C, Wood SN, Desta S, Galadanci H, Makumbi FE, et.al. Covert use of contraception in three sub Saharan African countries: A qualitative exploration of motivations and challenges. BMC Public Health. 2020; 20:865. https://doi.org/10.1186/s12889-020-08977-y

Adongo PB, Tapsoba P, Phillips JF, Tabong PT, Stone A, Kuffour E, et. al. The role of community-based health planning and services strategy in involving males in the provision of family planning services: A qualitative study in Southern Ghana. Reproductive Health. 2013; 10 (36): 1–15.

Kabagenyi A, Jennings L, Reid A, Nalwadda G, Ntozi J, Atuyambe L. Barriers to male involvement in contraceptive uptake and reproductive health services: a qualitative study of men and women's perceptions in two rural districts in Uganda. Reproductive Health. 2014; 11:1–9.

health clinic in a municipality in Ghana. BMC Womens Health. 2016;16(1)1–10.

Omitted Reference:

Roudsari RL, Farangis S, Goudarzi F. Barriers to the participation of men in reproductive health care: a systematic review and meta-synthesis. BMC Public Health. 2023; 23(818).https://doi.org/10.1186/s12889-023-15692-x

Ratcliffe AA, Hill AG, Walraven G. Separate lives, different interests: male and female reproduction in the Gambia. Bull World Health Organ. 2000;78(5):570–579.

Schuler S, Rottach E, Mukiri P. Gender norms and family planning decision-making in Tanzania: a qualitative study. Journal of Public Health in Africa. 2011;5:2:e25. DOI:10.4081/jphia.2011.e25. 19

Financial disclosure No changes

Reviewer 2:

The abstract is well-structured; however, consider summarizing the key findings in bold points for easier readability.

This had been amended, please see the abstract section.

Clearly define key terms early on, such as "family planning," "contraceptive use," and specific cultural terms related to fertility in Sudan

This has been amended; please see some changes in the introduction section.

The authors could strengthen the rationale for the study by explicitly stating the gaps in existing research and the significance of addressing these in the Sudanese context

This has been amended; please see the last section in the introduction.

The authors should have provided strong details on methodology; however, clarify the rationale behind the sample size (46 This has been mentioned in the data collection and trustworthiness sections of the study.

“There was no set limit to the number of interviews for the study. Saturation was reached when no new data emerged. Therefore, a total of 46 participants were included in this study.”

The decision to conduct interviews separately This has been mentioned in the data collection section.

“Drawing from the work modelled in Zipp and Toth [40, 41], the study paid close attention to factors that may shape or influence spouses' responses. To account for this, individual interviews were conducted on the same day with the same start and finish time, but always in separate spaces.”

Consider including additional information on any pilot testing done for interview guides or data collection methods to enhance credibility.

A summary of this is mentioned now in the data collections methods section. “Drawing upon literature on contraceptive decision making and the role of men and women in contraceptive decisions in Sub-Saharan Africa and other similar relevant contexts, questions were conceptualized but designed and formulated in accordance to specifically address the Sudanese context.

Annex 1 has a list of the interview guide questions for participants used in the study.

The four pathways are insightful. To enhance clarity, consider summarizing each pathway with a diagram or table that encapsulates the findings visually. Table added in the results section. S2 table.

When quoting participants, ensure a uniform presentation style to enhance readability.

• Tables (S1 and S2): Ensure that the tables are referenced consistently in the text. For instance, include contextual details before referencing a particular table.

Revised: This has been changed, please refer to the new, concise description of couples in brackets at the end of each quote, for easier readability

Discussion

• Integrate relevant theoretical frameworks that support your findings. This could enhance the academic level and context of your analysis. • Connect the findings to broader implications beyond Sudan, discussing how culturally influential decisions in family planning are a common issue in various regions globally. Changes have been made in the discussion to relate to broader findings and relevant theoretical frameworks

Your acknowledgment of limitations is good. It could be beneficial to offer suggestions for future research, highlighting how studies could be expanded or improved in method. An Additional title: ‘study implications’ is included now in the study

Conclusion

• The conclusion consolidates findings effectively. Reinforce the call to action for integrating men into family planning discussions, as it positions your research within a solution-focused frame. Changes are now made in the conclusion sections.

Ensure that all references are formatted consistently according to the selected citation style; this includes consistency in formatting year, volume, and pages in your references list. Reference list revised.

Reviewer 3:

The summary tables for the socio-demographic are not in the standard way. Revised.

The description of the qualitative approach is inadequate. The justification for the chosen sample size and the use of snowball sampling is weak, and these limitations should be acknowledged more transparently. This section has been revised AND included in the methods section. ‘The medical directors and key health services providers have extensive experience providing family planning and reproductive health services for many years in their respective communities and were, therefore, suitable to assist in identifying entry points and potential study participants in the health centres' catchment areas’.

This section has been included in the methods section:

‘Participant recruitment and inclusion criteria: Participants were selected using purposeful sampling from the communities served by the FRHC in Khartoum and the AFHC in Omdurman. Using a snowballing approach, the medical directors and key health services providers asked potential study participants to refer further participants. This snowballing approach was beneficial in finding and including relevant participants responding to the study's objective and ensuring the accumulation of information richness. *Patton MQ. Qualitative Evaluation and Research Methods. 1990. *Nyimbili F. Nyimbili L. Types of purposive sampling techniques with their examples and application in qualitative research studies. 2024.

The results section is still descriptive and lacks critical analysis. The themes identified from the interviews are support Revision made for a clearer analytic approach in all 4 pathways/themes

The discussion does not critically engage with the findings. It merely restates the results without offering deeper insights or considering alternative interpretations. The manuscript would benefit from a more critical engagement with the literature and a discussion of the broader implications of the findings.

Changes have been made in this section relate to broader findings and relevant theoretical frameworks.

---

## [Decision Letter · Decision Letter 3]

11 Dec 2025

Decisions and Choices about Fertility and Family Planning: Perspectives from Husbands and Wives in Sudan.

Dear Dr. Badri,

Thank you for submitting your manuscript to PLOS ONE. After careful consideration, we feel that it has merit but does not fully meet PLOS ONE’s publication criteria as it currently stands. Therefore, we invite you to submit a revised version of the manuscript that addresses the points raised during the review process.

I encourage you to address the comments raised by Reviewer #4 and make the necessary revisions. I look forward to reviewing your revised manuscript.

We look forward to receiving your revised manuscript.

Kind regards,

Godwin Banafo Akrong, Ph.D.

Academic Editor

PLOS One

Journal Requirements:

Reviewers' comments:

Reviewer's Responses to Questions

**Comments to the Author**

Reviewer #2: All comments have been addressed

Reviewer #4: (No Response)

2. Is the manuscript technically sound, and do the data support the conclusions?

Reviewer #2: Yes

Reviewer #4: Yes

3. Has the statistical analysis been performed appropriately and rigorously?

Reviewer #2: Yes

Reviewer #4: Yes

4. Have the authors made all data underlying the findings in their manuscript fully available?

Reviewer #2: Yes

Reviewer #4: Yes

5. Is the manuscript presented in an intelligible fashion and written in standard English?

Reviewer #2: Yes

Reviewer #4: Yes

Reviewer #2: This qualitative study offers valuable insights into the influence of husbands' fertility preferences on their wives' contraceptive use in Sudan, highlighting important societal and individual dynamics. However, there are several areas for improvement.

Strengths:

• The study addresses a critical reproductive health issue in Sudan, where low contraceptive use and high fertility remain challenges.

• Employing qualitative interviews provides depth and nuanced understanding of gender dynamics and personal beliefs.

• Including both husbands and wives in separate interviews allows for a comprehensive view of family planning decisions within relationships.

• The emphasis on policy and community engagement, including involving men in reproductive health discussions, is well-grounded.

Areas for Improvement:

• While 46 participants can provide rich qualitative data, the sample size and geographic focus on Khartoum and Omdurman may limit generalizability. Including rural areas or diverse socio-economic groups could enhance the comprehensiveness.

• Given the sensitive nature of family planning and gender roles, responses might be influenced by social desirability bias, especially when discussing male involvement and women’s agency. Strategies to mitigate this, such as ensuring confidentiality or triangulating data, should be explicitly discussed.

• The summary indicates common themes but lacks detail on how these themes emerged or how differences among participants were handled. A more detailed explanation of the coding process and thematic development would strengthen methodological transparency.

• Although some wives seek to assert their reproductive choices, the paper could delve deeper into the barriers they face when doing so, including cultural, social, or economic obstacles. This would provide a fuller picture of women’s agency.

• While recommendations are made for involving men and training health providers, specifics on how these strategies might be operationalized in the Sudanese context are limited. Including examples or frameworks could improve practical utility.

Reviewer #4: Comments

This topic is timely, given the increasing focus on how fertility preferences are shaped by demographic factors like education, economic status, and religious beliefs.

1. In the introduction, the author(s) should address the following:

a. Abbreviations should be provided in full before using them in the paper. E.g. ICPD

b. Also, perhaps it will be important for the author(s) to clearly define upfront some of the key terms they used in their study to ensure readers understand their specific meanings in the context of the study. E.g. family planning, contraceptive use, and fertility preferences.

2. The section title “aims” should be merged with the introduction.

3. The section titled “methodology” should be “methods.”

4. The colon placed in the subsections in the methodology section should be removed.

5. In using the snowballing approach, how did the author(s) address sample bias? This is because participants can be more inclined to recommend those who have similar traits or experiences, which could distort the sample.

6. What was the theoretical foundation of the study?

7. The author(s) make mention of data collectors; were these people trained? Who trained them?

8. S1 Table should be labelled Table 1.

9. In Table S1 Table, the column data collection methods should be deleted

10. In reporting the quotes made by participants, the author(s) should put them in this form “(Muslim- Wife, monogamous, three children)” rather than “(S1 Table. Couple: three children,

Khartoum).”

This is to help readers appreciate who is making the comments and the context in which they are being made.

11. The title states that the study will explore the perspective from husbands and wives, but the quotes reported as presented by the author(s) were only from the women. Can the author clarify why? Because it will be equally important to include the husbands' perspective?

12. The limitation section should be added to the discussion section.

13. Was the study shaped around any theoretical foundation?

14. Were the questions used for the interview guide adopted from a prior study?

15. With the interview guide design, were there any piloting? If yes, how was it carried out, and did it inform the final design of the interview guide?

**Do you want your identity to be public for this peer review?** For information about this choice, including consent withdrawal, please see our Privacy Policy

Reviewer #2: **Yes:** rehema abdallah

Reviewer #4: No

---

## [Author Response · Author response to Decision Letter 4]

23 Jan 2026

Date: 23 January 2026

To the Academic Editor PLoS One

Subject: Re-submitting the revised manuscript. ‘Decisions and Choices about Fertility and Family Planning: Perspectives from Husbands and Wives in Sudan’.

Thank you for allowing me to resubmit the revised manuscript, inspired by your esteemed journal's article on the practical implications of cultural and social aspects in reproductive health-related issues.

Please find a response to the Academic editor, reviewers 2 and 4 noted below.

Academic Editor:

Please review your reference list. Changes to the reference list should be mentioned in the rebuttal letter

The text and the reference list have been revised accordingly. The additions and omitted references below are better suited to the revised manuscript.

Added references:

Schuler S, Rottach E, Mukiri P. Gender norms and family planning decision-making in Tanzania: a qualitative study. Journal of Public Health in Africa. 2011;5:2:e25. DOI:10.4081/jphia.2011.e25. 19

Comerasamy H, Read B, Francis C, Cullings S, Gordon H. The acceptability and use of contraception: A prospective study of Somalian women’s attitudes. Journal of Obstetrics and Gynaecology. 2003; 23 (4): 412–415

Atkinson R, Flint J. Accessing hidden and hard-to-reach populations: Snowball research strategies. Social Research Update. 2001

Liebenberg L, Jamal A, Ikeda J. Extending youth voices in a participatory thematic analysis approach. International Journal of Qualitative Methods. 2020; 19, 1609406920934614. https://doi.org/10.1177/1609406920934614

Corbin J, Strauss A. Basics of qualitative research (3rd ed.): Techniques and procedures for developing grounded theory. Sage. 2012.

Miles MB, Huberman AM, Saldanã J. Qualitative data analysis: A methods sourcebook (4th ed.) [Kindle edition]. Sage. 2020.

Akindele RA, Adebimpe WO. Encouraging male involvement in sexual and reproductive health: family planning service providers’ perspectives. Int J Reprod Contracept Obstet Gynecol. 2013;2:119–23. 32.

Kwambai TK, Dellicour S, Desai M, Ameh CA, Person B, Achieng F, et al. Perspectives of men on antenatal and delivery care service utilisation in rural western Kenya: a qualitative study. BMC Pregnancy Childbirth. 2013;13. Available from: http://www.biomedcentral.com/1471-2393/13/134

Shaa El deen M, Al Rasheed F. Reproductive health and causes of infertility among men and women in Sudan. Documentary Program, Balad Fe Shasha [A Country’s View]. Sudan Television Channel [S24]. 2023.

Letshwenyo-Maruatona S. Who do Batswana men prefer: male or female health providers?. American Journal of Men’s Health. 2017; 11(6) 1642–1652. DOI: 10.1177/1557988315621727

Omitted Reference:

Mahfouz MS. Fertility in Northern Sudan (1979-1999): levels, trends and determinants. Paper presented at the XXVI International Population Conference of the IUSSP Morocco. 2009.

Patton MQ. Qualitative Evaluation and Research Methods (2nd ed.). Newbury Park, CA: Sage. 1990.

Mkandawire P, MacPherson K, Madut K, Odwa DA, Rishworth A, Luginaah I. Men’s perceptions of women’s reproductive health in South Sudan. Health and Place 58. 2019.

Reviewer 2:

1.While 46 participants can provide rich qualitative data, the sample size and geographic focus on Khartoum and Omdurman may limit generalizability. Including rural areas or diverse socio-economic groups could enhance the comprehensiveness The following is amended in the Methods section: p. 6 & 7. And limitations section: p.18

2.Given the sensitive nature of family planning and gender roles, responses might be influenced by social desirability bias, especially when discussing male involvement and women’s agency. Strategies to mitigate this, such as ensuring confidentiality or triangulating data, should be explicitly discussed.

This has been amended in the methods section: p. 7 & p. 8

3. The summary indicates common themes but lacks detail on how these themes emerged or how differences among participants were handled. A more detailed explanation of the coding process and thematic development would strengthen methodological transparency. This has been amended in the methods section. p. 8

4.Although some wives seek to assert their reproductive choices, the paper could delve deeper into the barriers they face when doing so, including cultural, social, or economic obstacles. This would provide a fuller picture of women’s agency This has been amended in the introduction section. p. 3

5. While recommendations are made for involving men and training health providers, specifics on how these strategies might be operationalized in the Sudanese context are limited. Including examples or frameworks could improve practical utility.

This has been amended in the study implications & conclusion sections: p. 18 & 19

Reviewer 4:

1.In the introduction, the author(s) should address the following:

a. Abbreviations should be provided in full before using them in the paper. E.g. ICPD

b. Also, perhaps it will be important for the author(s) to clearly define upfront some of the key terms they used in their study to ensure readers understand their specific meanings in the context of the study. E.g. family planning, contraceptive use, and fertility preferences.. Pt. a is amended in introduction section: p. 3

Pt. b is amended in introduction section: p. 5

2.The section title “aims” should be merged with the introduction.

This has been amended in introduction section: p. 4

3. The section titled “methodology” should be “methods.” This has been amended in methods section: p. 5

4.The colon placed in the subsections in the methodology section should be removed. This has been amended in methods section: p. 5-8

5.In using the snowballing approach, how did the author(s) address sample bias? This is because participants can be more inclined to recommend those who have similar traits or experiences, which could distort the sample.

This has been amended in methods section: under sub heading Participant recruitment and inclusion criteria p. 5

6.What was the theoretical foundation of the study? This has been amended in methods section: under sub heading Trustworthiness of the study. p.8

7.The author(s) make mention of data collectors; were these people trained? Who trained them?

This has been amended in methods section: under sub heading data collection & study procedure p. 6

8.S1 Table should be labelled Table 1. This has been amended in methods section p. 8 and (supporting files table 1)

9.In Table S1 Table, the column data collection methods should be deleted This has been amended in results section p. 9 and (supporting files table 1)

10.In reporting the quotes made by participants, the author(s) should put them in this form “(Muslim- Wife, monogamous, three children)” rather than “(S1 Table. Couple: three children,

Khartoum).”

This is to help readers appreciate who is making the comments and the context in which they are being made.

This has been amended in results section: p. 11-16

11. The title states that the study will explore the perspective from husbands and wives, but the quotes reported as presented by the author(s) were only from the women. Can the author clarify why? Because it will be equally important to include the husbands' perspective?

Kindly refer to results section: pathways

1-3 p. 11-13.

The quotes indicate the couple's response (husband & their wife)

12. The limitation section should be added to the discussion section This has been amended in discussion section: p. 17

13.Were the questions used for the interview guide adopted from a prior study?

This has been amended in methods section: under sub heading Interview guide p. 7

14.With the interview guide design, were there any piloting? If yes, how was it carried out, and did it inform the final design of the interview guide? a. Kindly be informed that this study did not have a pilot study prior to commencement. However, as noted in the manuscript (methods section), several steps were taken to facilitate data collection and ensure methodological rigour.

Please refer to methods section: sub heading Data collection and study procedure p. 7 for details of facilitation of data collection. for e.g. same gender data collectors for participants, conducting interviews with couples in separate spaces etc.

b. Further, the convening of frequent meetings, reviewing work progress, discussing preliminary findings, and discussing any emerging themes were important to validate data. Please

prefer to methods section subheading Data collection and study procedure p. 6

Points a & b are summarised in methods section: sub heading Trustworthiness of the study p. 8

---

## [Decision Letter · Decision Letter 4]

12 Feb 2026

Decisions and Choices about Fertility and Family Planning: Perspectives from Husbands and Wives in Sudan.

PONE-D-24-10658R4

Dear Dr. Badri,

We’re pleased to inform you that your manuscript has been judged scientifically suitable for publication and will be formally accepted for publication once it meets all outstanding technical requirements.

Kind regards,

Godwin Banafo Akrong, Ph.D.

Academic Editor

PLOS One

Additional Editor Comments (optional):

Reviewers' comments:

Reviewer's Responses to Questions

**Comments to the Author**

Reviewer #4: All comments have been addressed

2. Is the manuscript technically sound, and do the data support the conclusions?

Reviewer #4: Yes

3. Has the statistical analysis been performed appropriately and rigorously?

Reviewer #4: Yes

4. Have the authors made all data underlying the findings in their manuscript fully available?

Reviewer #4: Yes

5. Is the manuscript presented in an intelligible fashion and written in standard English?

Reviewer #4: Yes

Reviewer #4: I don't have any additional comments. Authors have responded and worked on my suggested changes/comments

**Do you want your identity to be public for this peer review?** For information about this choice, including consent withdrawal, please see our Privacy Policy

Reviewer #4: No

---

## [Editor Report · Acceptance letter]

PONE-D-24-10658R4

PLOS One

Dear Dr. Badri,

I'm pleased to inform you that your manuscript has been deemed suitable for publication in PLOS One. Congratulations! Your manuscript is now being handed over to our production team.

Kind regards,

on behalf of

Dr. Godwin Banafo Akrong

Academic Editor

PLOS One